# Fabrication and Characterization of a Self-Powered n-Bi_2_Se_3_/p-Si Nanowire Bulk Heterojunction Broadband Photodetector

**DOI:** 10.3390/nano12111824

**Published:** 2022-05-26

**Authors:** Xuan Wang, Yehua Tang, Wanping Wang, Hao Zhao, Yanling Song, Chaoyang Kang, Kefan Wang

**Affiliations:** 1Henan Province Key Laboratory of Photovoltaic Materials, Henan University, Kaifeng 475004, China; 104754190988@henu.edu.cn (X.W.); 80020012@vip.henu.edu.cn (Y.T.); 104754191019@henu.edu.cn (W.W.); songyl@163.com (Y.S.); 2School of Physics and Electronics, Henan University, Kaifeng 475004, China; 18211973341@163.com

**Keywords:** topological insulator, Bi_2_Se_3_, silicon nanowires, photodetector, bulk heterojunction

## Abstract

In the present study, vacuum evaporation method is used to deposit Bi_2_Se_3_ film onto Si nanowires (NWs) to form bulk heterojunction for the first time. Its photodetector is self-powered, its detection wavelength ranges from 390 nm to 1700 nm and its responsivity reaches its highest value of 84.3 mA/W at 390 nm. In comparison to other Bi_2_Se_3_/Si photodetectors previously reported, its infrared detection length is the second longest and its response speed is the third fastest. Before the fabrication of the photodetector, we optimized the growth parameter of the Bi_2_Se_3_ film and the best Bi_2_Se_3_ film with atomic steps could finally be achieved. The electrical property measurement conducted by the physical property measurement system (PPMS) showed that the grown Bi_2_Se_3_ film was n-type conductive and had unique topological insulator properties, such as a metallic state, weak anti-localization (WAL) and linear magnetic resistance (LMR). Subsequently, we fabricated Si NWs by the metal-assisted chemical etching (MACE) method. The interspace between Si NWs and the height of Si NWs could be tuned by Ag deposition and chemical etching times, respectively. Finally, Si NWs fabricated with the Ag deposition time of 60 s and the etching time of 10 min was covered by the best Bi_2_Se_3_ film to be processed for the photodetector. The primary n-Bi_2_Se_3_/p-Si NWs photodetector that we fabricated can work in a self-powered mode and it has a broadband detection range and fast response speed, which indicates that it can serve as a promising silicon-based near- and mid-infrared photodetector.

## 1. Introduction

With the rapid development of the semiconductor industry, semiconductor photodetectors are widely used in the military, for scientific research and in daily life. In order to promote the performance of photodetectors, many revolutionary materials are chosen to fabricate them. Specially, two-dimensional (2D) materials, such as graphene [1] and transition metal dichalcogenides (TMDCs) [2,3], are considered as promising photodetection materials due to their ultra-high carrier mobility, superior integrability and broadband light absorption properties. However, the practical application of a graphene photodetector has been limited by its zero bandgap, which results in an extremely high dark current [1]. The TMDCs photodetectors are limited by their low detectivity of infrared light and complex fabrication techniques [2,3].

The topological insulator (TI) is another kind of novel material with internal insulation and a gapless surface [4,5]. The carriers on its surface are protected from backscattering due to the symmetry of time inversion. It also presents strange physical phenomena, such as spin–orbital coupling, quantum oscillation and weak anti-localization (WAL) effect [5,6,7,8,9]. Therefore, it is widely used in spintronic devices, optoelectronic devices, flexible electrodes and quantum computing [10,11,12,13]. Bi_2_Se_3_ is the first certified material as the second kind of three-dimensional TI, which has the advantage of simple surface state structure. In addition, its direct bandgap is ~0.3 eV, which makes it less be susceptible to room-temperature thermal disturbance and can absorb light with wavelengths of up to ~4 μm [14]. These characteristics give Bi_2_Se_3_ the potential to prepare high-performance infrared photoelectric detectors [15].

On the other hand, silicon material has been widely used in the fields of optoelectronic and microelectronic devices. However, it has an indirect bandgap of 1.12 eV at room temperature, which means that it cannot detect infrared light with a wavelength >1.1 μm, including the two important communication wavelengths 1.3 μm and 1.55 μm. In order to extend the silicon detection wavelength, Bi_2_Se_3_ has been used to deposit onto silicon to form a heterojunction infrared detector [16,17,18,19]. Hongbin Zhang et al. [16] fabricated Bi_2_Se_3_ by the physical vapor deposition (PVD) method inside a horizontal tube furnace, and the Bi_2_Se_3_/n-Si photodetector with a detection range of 350 nm to 1100 nm, a responsivity of 24.28 A/W, a detectivity of 4.39 × 10^12^ Jones, a rise time (t_r_) of 2.5 μs and a fall time (t_f_) of 5.5 μs. Additionally, Xin Hong et al. [17] fabricated Bi_2_Se_3_ film on a micropyramidal Si substrate also by the PVD method inside a tube furnace, and the photodetector had a response wavelength range of 635 to 2700 nm, a photo-to-dark ratio of 1.01 × 10^4^, a dark current of 0.11 nA, a t_r_ of 0.52 ms and a t_f_ of 0.44 ms. Liu C et al. [18] fabricated Bi_2_Se_3_ nanowires on the n-Si substrate by an Au-catalyzed vapor–liquid–solid (VLS) method, and its photodetector had a response wavelength range of 380 to 1310 nm, a photo-to-dark ratio of 10, a responsivity of 924.2 A/W, a t_r_ of 45 ms and a t_f_ of 47 ms. Since Bi_2_Se_3_ is an intrinsic n-type material, we believe that n-Bi_2_Se_3_/p-silicon heterojunction is more favorable for electron transportation [19]. Mingze Li et al. [19] fabricated a Cu-doped n-Bi_2_Se_3_ film on a p-Si substrate by the chemical vapor deposition (CVD) method inside a high-temperature tube furnace, and its photodetector possessed a detection range of 400 nm to 1200 nm and a responsivity of 168.9 mA/W, a t_r_ of 4 ms and a t_f_ of 4 ms. Xujie Pan et al. [20] fabricated a Bi_2_Se_3_ film on an Si (111) substrate by the molecular beam epitaxy (MBE) method, and its photodetector had a response of 800 nm to 1200 nm, a t_r_ of 2 ms and a t_f_ of 2 ms. Biswajit Das et al. [21] used the chemical method to synthesize the Bi_2_Se_3_ nanoflake (NF) on the Si NWs surface and fabricated a Bi_2_Se_3_ NF/Si NWs heterojunction photodetector, which had a detection wavelength range of 300 nm to 1000 nm, a responsivity of 934.1 A/W, a detectivity of 2.30 × 10^13^ Jones, a t_r_ of 41 ms and a t_f_ of 79 ms.

To date, however, the vacuum evaporation technique has not been used to deposit Bi_2_Se_3_ films for a photodetector. In comparison to MBE [20,22], its apparatus is much cheaper and easier to maintain. In comparison to PVD and CVD in a tube furnace or chemical synthesis [16,19,23], it can deposit a large area and uniform film with good reproducibility, and is easier to incorporate into an advanced Si integrated circuit (IC) technology. In order to extend the response wavelength and the responsivity of a Bi_2_Se_3_ photodetector at the mid-infrared (MIR) zone, a thick (>100 nm) Bi_2_Se_3_ film with better IR absorption is needed [24]. However, the lifetime of a minority carrier in Bi_2_Se_3_ is only 50 ps at 300 K [25], and thus the diffusion length of the minority carrier would also be very small. In order to facilitate the collection of a photo-generated carrier and reduce the recombination, in the present study, we plan to deposit a thick Bi_2_Se_3_ film of 100 nm onto Si NWs to form bulk heterojunction to shorten the diffuse distance of minority carriers to the dissociating interface. In the bulk heterostructure, two materials form interpenetrating networks minimize the distance that excitons must travel before dissociating at a heterointerface [26,27]. As a result, the photo-generated carriers are efficiently collected before they are recombined. In addition, Si NWs can be simply fabricated by the MACE method and its morphology can be easily tuned [28,29,30].

In this paper, Bi_2_Se_3_ film was grown by the vacuum thermal evaporation method and its growth parameters were optimized to obtain the best growth condition. The optical and electrical properties of the best Bi_2_Se_3_ film were investigated. Then, the Si NWs were grown by the MACE method and their interspace and height were tuned by the Ag deposition time and etching time. Finally, the performance of the best n-Bi_2_Se_3_/p-Si NWs photodetector was measured and analyzed.

## 2. Materials and Methods

### 2.1. Sample Preparation

#### 2.1.1. Preparation of Bi_2_Se_3_ Thin Films

The crystalline Bi_2_Se_3_ film was deposited on an Si (111) substrate by the vacuum thermal evaporation method. The background vacuum was lower than 3 × 10^−5^ Pa. Firstly, a p-type <111> single crystal silicon wafer (with a resistivity of 1–10 Ωcm^−1^) was cleaned following the standard RCA process. Before it was introduced into the vacuum growth chamber, it was dipped into a hydrofluoric acid (HF) buffer solution (HF:C_2_H_5_OH = 1:10 (*v*/*v*)) to create a H-passivated Si (111) surface. This Si substrate was degassed at 200 °C for 30 min and then it was heated to the growth temperature and kept at this temperature for 2 h to reach the thermal equilibrium before the Bi_2_Se_3_ growth. High-purity Bi and Se granules (both bought from Zhongnuo Advanced Material (Beijing, China) Technology with a purity of 99.999%) were co-evaporated from two separate Knudsen effusion cells. A series of experiments were conducted at different growth temperatures (220~420 °C) and different ratios of Bi/Se evaporation rates. The evaporation rates of the Bi and Se sources were controlled by their effusion cell temperatures. Following the Bi_2_Se_3_ depositions, the temperature of the sample holder was cooled down naturally at an average rate of 1 °C/min.

#### 2.1.2. Preparation of Si NWs

The MACE method was used to fabricate the Si NWs by using Ag nanoparticles (NPs) as the catalyst. The P-type double-polished Si (100) wafer with a resistivity of 1–10 Ωcm^−1^ was used as the start material. Firstly, the Si (100) wafer was put into the water solution mixture of HF (5 mol/L) acid and AgNO_3_ (0.01 mol/L) to deposition Ag NPs at the sample surface. Secondly, the sample was put into the water solution mixture of HF (5 mol/L) acid and H_2_O_2_ (0.2 mol/L) to etch the silicon surface to produce the Si NWs. Thirdly, the HNO_3_ acid (6.54 mol/L) water solution was used to remove the remain Ag NPs inside the Si NWs. The height and density of Si NWs can be controlled by the etching time and diameter (deposition time) of Ag NPs, respectively. Lastly, the Si NWs samples were dried in a drying oven and were then ready for characterization. 

#### 2.1.3. Fabrication of the n-Bi_2_Se_3_/p-Si NWs Photodetector

The prepared Si NWs samples were used as the substrate for the Bi_2_Se_3_ film deposition. Then, an interdigital Ag film was deposited on the Bi_2_Se_3_ surface as the front-side metal electrode and a fully covered Ag film was deposited onto the Si surface as the backside metal electrode. Both of the Ag films were deposited by magnetron sputtering. The purity of the Ag piece was 99.99% and high-purity (5N) Ar gas was introduced at a rate of 50 sccm. The work pressure of the vacuum system was stabilized at around 1.4 × 10^−2^ mbar. The power of the radio-frequency (RF) source was 60 W. The deposition thickness of the Ag films was 60 nm. 

### 2.2. Sample and Device Testing Method

The Bi_2_Se_3_ thin films were characterized by X-ray diffraction (XRD, D8 Advance, Bruker, Berlin, Germany), atomic force microscope (AFM, Multimode 8, Bruker, Berlin, Germany), Raman (inVia, Renishaw, New Mills, UK), X-ray photoelectron spectroscopy (XPS, AXIS, SUPRA+, Shimadzu, Milton Keynes, UK), scanning electron microscope (SEM, JEM-F200, JEOL, Tokyo, Japan) and PPMS (9 Ever Cool II, Quantum Design, San Diego, CA, USA). The Si NWs and the Bi_2_Se_3_/Si NWs were characterized by SEM (JEM-F200, JEOL, Tokyo, Japan) and UV-VIS-NIR Spectroscopy (Lambda 950, PerkinElmer, Akron, OH, USA). The performance of the device was tested by a semiconductor characteristics system (4200-SCS, Keithley, Beaverton, OR, USA) and a spectral response test system, which consisted of the following parts: light source (model 7ILT250 halogen tungsten light source with a power of 250 W, Sofn Instruments, Beijing, China), chopper (model 3501 Optical Chopper, Newport, Andover, MA, USA), monochromator (model 7ISW151 dual raster scanning spectrometer, Sofn Instruments, Beijing, China), lock-in amplifier (model SR830 DSP, Stanford Research Systems, Sunnyvale, CA, USA), DC power supply, one-dimensional lift table for sample testing, standard detector (model DET 36A, Thorlabs, Newton, NJ, USA), optical path components and related test software [31].

## 3. Results and Discussion

### 3.1. Growth and Characterization of Bi_2_Se_3_ Films

In order to obtain the optimal growth conditions for the Bi_2_Se_3_ film, we deposited the Bi_2_Se_3_ film on the Si (111) surface at different temperatures of the sample holder and different Bi/Se beam ratios. The Bi/Se beam ratios were realized by controlling the evaporation temperatures of Bi and Se source cells. The crystal phase, atomic vibration and surface morphology of the Bi_2_Se_3_ film were characterized by XRD, Raman and AFM, respectively. When the temperatures of the Se cell and sample holder were fixed at 209 °C and 320 °C, respectively, we deposited the Bi_2_Se_3_ film on the Si (111) substrate at different Bi cell temperatures from 650 to 850 °C with an interval temperature of 50 °C. The XRD, Raman and AFM of the obtained Bi_2_Se_3_ films are shown in Figure 1. From Figure 1a, we can observe that, at the Bi cell temperatures of 800 °C and 850 °C, the stoichiometric Bi/Se ratio inside the obtained BiSe alloy film is greater than that in Bi_2_Se_3_. When the Bi cell temperature is at 700 °C, the diffraction peaks of the Bi_2_Se_3_ film are very weak. For the Bi cell temperature of 650 °C, no diffraction exists. When the Bi cell temperature is 750 °C, however, the diffraction peaks of Bi_2_Se_3_ are very strong and they are at 9.34°, 18.6°, 37.52°, 47.44°, 57.68° and 68.51°, corresponding to the (003), (006), (0012), (0015), (0018) and (0021) planes of the Bi_2_Se_3_ film. Only the diffraction peaks of the (00N) plane appear in Figure 1a, which indicates that the Bi_2_Se_3_ film grows strictly in the direction of the c axis [32,33]. 

Figure 1b shows the Raman spectroscopy of the Bi_2_Se_3_ films. For the Bi_2_Se_3_ film grown at a Bi cell temperature of 750 °C, we can observe three distinct Raman shift peaks at 73 cm^−1^, 131 cm^−1^ and 174 cm^−1^, which can be assigned to A_1g_^1^, E_g_^2^ and A_1g_^2^, respectively [34]. As is well known, the topological insulator Bi_2_Se_3_ is a 2D material. Each of its units is composed of five layers of atoms and can be denoted as the Se^I^, Bi, Se^II^, Bi and Se^I^ layers. The A_1g_ vibrational mode is caused by the out-of-plane symmetric stretching in the same direction of the outer pair of atoms (Se^I^-Bi) or (Bi-Se^I^). When the outer pair of atoms, (Se^I^-Bi) or (Bi-Se^I^), stretch symmetrically in the opposite direction, the A_1g_^2^ vibrational mode is produced. E_g_^2^ is caused by the in-plane symmetric stretching of the outer layer atoms (Se^I^-Bi) or (Bi-Se^I^). During these atomic vibration modes, the central Se^II^ atomic layer always remains stationary [35,36]. For the BiSe sample grown with Bi cell temperatures of 800 °C or 850 °C, many vibrational modes induced from other Bi_x_Se_y_ alloys appeared. For the Bi cell temperature of 700 °C, the three vibrational modes were weaker than those at 750 °C, indicating the thinner film of Bi_2_Se_3_ and thus the lower deposition rate. According to the previous literature [37], the growth rate of Bi_2_Se_3_ film depends on the Bi deposition rate in an Se-rich atmosphere (evaporation rate: Se:Bi = ~10:1). The Bi deposition rate depends on the temperature of the Bi source cell. When the Bi cell temperature is 700 °C, the XRD diffraction peak of the Bi_2_Se_3_ film cannot be observed in Figure 1a, but its weak Raman peaks can be observed in Figure 1b. These findings indicate that the thickness of the deposited Bi_2_Se_3_ film is very thin. For the Bi cell temperature of 650 °C, however, none of the three characteristic vibrational modes of Bi_2_Se_3_ exist, which demonstrates that no Bi_2_Se_3_ film was deposited on the Si surface, and thus no Bi was evaporated at this cell temperature. In general, the Raman results for the Bi_2_Se_3_ film are consistent with their XRD results presented in Figure 1a.

Figure 1c–g show the surface morphology of Bi_2_Se_3_ film grown with different Bi cell temperatures. Only at 750 °C did the Bi_2_Se_3_ film have the obvious angularity of crystalline grain and the atomic layers, which indicates that the Bi_2_Se_3_ film follows a layer-by-layer growth mode in this grow condition. At the Bi cell temperatures of 800 and 850 °C, as shown in Figure 1f,g, a large size of agglomerate BiSe NPs exist. At the Bi cell temperature of 650 °C, we can observe from Figure 1a,b that there is no Bi_2_Se_3_ at the film surface. Therefore, the NPs should be some remaining Se. At the Bi cell temperature of 700 °C, the surface should exhibit a few Bi_2_Se_3_ NPs. 

Following the above optimization, we discovered that the optimal cell temperature for Bi evaporation was 750 °C. Then, we fixed the temperature at 750 °C and tuned the Se cell temperature from 194 °C to 219 °C at a temperature interval of 5 °C. The XRD, Raman and AFM results of BiSe films are shown in Appendix A (see the Appendix A). Appendix A shows that the deposited BiSe films are all in composition of Bi_2_Se_3_. The strongest of diffraction peak comes from the sample grown at the Bi cell temperature of 209 °C, which demonstrates that this Bi_2_Se_3_ film has the best crystallinity. The Raman results in Appendix A show that the strongest Raman peak also comes from the Bi_2_Se_3_ film grown at an Se cell temperature of 209 °C, consistent with the results presented in Appendix A. Appendix A show that only the Bi_2_Se_3_ film grown in Se cell temperatures of 204 °C or 209 °C have the atomic step at the film surface, and thus should express a layer-by-layer growth mode. This illustrates that, when the Bi and Se cell temperatures are 750 °C and 204 °C (or 209 °C), respectively, they have the best Bi/Se beam ratio to deposit the 2D materials, Bi_2_Se_3_ film. Note that, in Appendix A, there are many triangle Bi_2_Se_3_ crystalline islands, which agrees well with the triangle atomic arrangement at the Si (111) surface, and so indicates that the Bi_2_Se_3_ film has a good epitaxy relationship with the underneath Si substrate.

Subsequently, we fixed the Bi cell temperature at 750 °C and Se cell temperature at 209 °C, and then began to optimize the growth temperature for the Bi_2_Se_3_ film. The growth temperature was changed from 220 °C to 420 °C in a temperature interval of 50 °C. Appendix A show that the Bi_2_Se_3_ sample grown at 270 °C has the highest diffraction and vibration peaks, which indicates that it has the best crystallinity. Appendix A show that there are many atomic steps on the surface of the Bi_2_Se_3_ films grown at 220 °C, 270 °C, 320 °C and 370 °C. For the film grown at 420 °C, there is no Bi_2_Se_3_ on the surface, as shown in Appendix A, possibly due to the growth temperature that is too high to absorb the arriving Se atom.

Following the growth optimization, we deposited an optimal Bi_2_Se_3_ film sample with a Bi cell temperature of 750 °C, an Se cell temperature of 209 °C and a growth temperature of 270 °C. The deposition time was 15 min and the sample thickness was 30 nm. Its XRD, Raman, XPS and AFM results are shown in Figure 2. The XRD and Raman results determine that the Bi_2_Se_3_ film has a good purity and crystallinity. Figure 2c,d show the XPS spectra of Bi 4f and Se 3d, respectively. The Bi 4f spectrum was fitted with Bi 4f_7/2_ and Bi 4f_5/2_ peaks and present at binding energies of 158 and 163.3 eV, respectively. The weak fitting peaks centered at 159.1 and 164.25 eV showed a slight oxidation of the Bi_2_Se_3_ film [38]. The Se 3d spectrum can be perfectly fitted by Se 3d_5/2_ and Se 3d_3/2_ peaks present at binding energies of 53.5 and 54.4 eV, respectively [39]. The integral areas of Bi 4f and Se 3d showed an Se atomic percentage of 57.6% and a Bi atomic percentage of 42.4%, corresponding to an Se/Bi atomic ratio of 1.36. This is consistent with the EDS analysis shown in Appendix A and again proves a lack of Se in the Bi_2_Se_3_ film. Figure 2e shows that, beside the triangle Bi_2_Se_3_ island, there are many atomic steps on the sample’s surface. Then, we zoomed into a small zone (circled by a red rectangle in Figure 2e) in the AFM and measured its profile along the white line, as shown in Figure 2f. The atomic step profile is shown in Figure 2g. Two atomic steps with a height of 1 nm can be observed, of which the value agrees well with the previous literature [40,41]. That is to say, we can obtain the 2D materials, Bi_2_Se_3_ film, at the optimal growth condition.

In order to further investigate the chemical composition of the optimal Bi_2_Se_3_ film, we performed an SEM measurement. Appendix A shows the SEM of the sample surface. Appendix A shows the elemental distribution of the Bi_2_Se_3_ film. We can observe that the Bi and Se elements distribute very uniformly. Appendix A presents that the atomic percentages of Se and Bi are 59.39% and 40.61%, respectively. Their atomic ratio was 1.46, less than the stoichiometric ratio of 1.5 in Bi_2_Se_3_, possibly due to the effumability of the Se atom (melting point: 217 °C) at a growth temperature of 270 °C [21].

### 3.2. Analysis of Electrical Properties of the Samples

In order to avoid the effect of the Si substrate on the electrical transport property of the Bi_2_Se_3_ film, we deposited the Bi_2_Se_3_ film at the optimal growth condition on the insulative SrTiO_3_ (STO) substrate. The schematic of the sample and electrode is shown in Figure 3a. Figure 3b shows the curve of the longitudinal resistance (R_xx_) of the sample as a function of temperature. The resistance of the sample increased as the temperature increased from 13 K to 300 K, indicating the metal properties of the sample, which is consistent with the results reported in the literature [42]. However, between 3 K and 13 K, the resistance increased as the temperature decreased. It may be attributed to the strong correlation between electrons at low temperatures, which causes a Mott gap, leading to the metal-insulation transition [43,44]. Figure 3c shows that the Hall resistance of the sample (R_xy_) changes linearly with the magnetic field in the temperature range of 2~100 K, and its slope is negative, indicating that the Bi_2_Se_3_ is in the form of n-type conductivity, which may be due to Se vacancies.

Figure 3d shows the variation of the normalized magnetoresistance (MR) with the magnetic field of the sample in the temperature range of 2~100 K. The definition of MR is MR = [R(B) − R (B = 0)]/R (B = 0) × 100% [45]. The MR slowly changes between 0 T and 3 T, and increases linearly between 3 T and 9 T. This LMR is related to the scattering of massless Dirac fermions on the surface and also to the gapless surface states of TI. Abrikosov proposed a model based on quantum MR, which can explain the LMR effect when the magnetic field is strong enough that all Dirac fermions are quantized to the lowest Landau level [46]. The unusual LMR suggests that our thin film may be characterized by gapless surface states and Dirac fermions. It is worth noting that, at 10 K and below, the MR of the sample sharply increases near 0 T, which is caused by the WAL effect of TIs [47]. When the temperature exceeds 10 K, the change in the trend of MR has a quadratic dependence with B; that is, a parabolic MR appears, which is mainly due to the weakening of the WAL effect and the Lorentzian deflection of the carriers [48]. The above results show that the Bi_2_Se_3_ films grown on STO substrates have the general properties of topological insulators, also indicating the high-quality growth of the Bi_2_Se_3_ films. 

### 3.3. Results Analysis of Bi_2_Se_3_/Si NWs Heterojunction

If depositing the Bi_2_Se_3_ film on the Si NWs’ surface, it can enter the interspace between the NWs and form the bulk heterojunction optoelectronic device. The bulk heterojunction can usually facilitate the separation and transportation of the photon-generated carriers. Firstly, we changed the deposition time of Ag NPs to tune the interspace between Si NWs. With the different reaction times from 20 s to 100 s, the AgNO_3_ + HF solution deposits different sizes and densities of Ag NPs, as shown in Appendix A. It can be observed that the surface coverage of Ag NPs continuously increases with the deposition time. Then, the Ag NPs samples were put into HF+H_2_O_2_ solution to etch 10 min. The sample surface and cross-section morphology of the Si NWs are shown in Appendix A, respectively. Naturally, we can observe that the interspace between Si NWs increases with the deposition time of Ag NPs. Cross-section SEM verifies this increase in interspace with the Ag NPs deposition time. Secondly, we changed the etching time to tune the height of Si NWs. The surface and cross-section morphology of Si NWs with different etching times from 4 to 14 min and the same Ag NPs deposition time of 60 s are shown in Appendix A, respectively. We can observe that the Si NWs have the same densities, but with different heights from 1.2 µm to 4.7 µm. According to Appendix A, we plotted the height of Si NWs as a function of the etching time, as shown in Appendix A. The height of Si NWs continuously increases with the etching time.

Then, we deposited the 100 nm thick Bi_2_Se_3_ films on the Si NWs’ surface. Appendix A show the surface and cross-section SEM of the Bi_2_Se_3_/Si NWs samples with the same etching time of 10 min, but the different deposition time of Ag NPs from 20 s to 100 s. From Appendix A, we can observe that the interspace between Si NWs increases with the Ag deposition time. As the interface increases, a part of the Bi_2_Se_3_ particles enters the interspace increase and the part on the surface of the Si NWs simultaneously decreases, as shown in Appendix A. From Appendix A, we can observe that the part of Bi_2_Se_3_ particles on the surface of Si NWs is large enough to form a continuous film for Ag deposition times from 20 s to 60 s, as shown in Appendix A, but there are many apertures inside the surface Bi_2_Se_3_ film for the Ag deposition time of 100 s, as shown in Appendix A. In the meantime, we can find from the SEM that the surface roughness increases qualitatively with the Ag deposition time. Appendix A show the surface and cross-section SEM of the Bi_2_Se_3_/Si NWs samples with the same deposition times (60 s) of Ag NPs, but different etching times of 4 min to 14 min. Similarly, the interspace between Si NWs also slightly increases with the etching time, as shown in Appendix A. Therefore, the Bi_2_Se_3_ film on top of Si NWs has many apertures for the longer etching time, such as 12 min and 14 min, as shown in Appendix A. In the meantime, we can find from the SEM that the surface roughness increases qualitatively with the etching time. For the fabrication of the Bi_2_Se_3_/Si NWs photodetector, a part of the Bi_2_Se_3_ film needs to enter the interface of Si NWs to form the bulk heterojunction, and part of them should stay on the surface of the Si NWs to support the formation of the metal electrode. Finally, we chose the Bi_2_Se_3_/Si NWs sample with an Ag deposition time of 60 s and an etching time of 10 min to process the photodetector. For the sake of simplicity, next, we only show its morphology, optical absorption and device performance.

Figure 4a,b show the surface and cross-section SEM of Si NWs with the deposition time of 60 s for Ag NPs and the etching time of 10 min, respectively. It can be observed that the interspace between the Si NWs is uniform and also large enough for subsequent Bi_2_Se_3_ entering to form the bulk heterojunction. The height of Si NWs is about 2.8 µm. Following the Bi_2_Se_3_ deposition, the sample’s surface and cross-section SEM images are shown in Figure 4c,d. The surface of Si NWs is covered with Bi_2_Se_3_ nanosheets deposited at the optimal growth condition. The randomly distributed Bi_2_Se_3_ nanosheet has a diameter of ~450 nm and a thickness of ~70 nm. Figure 4d shows that the Bi_2_Se_3_ enters into the Si NWs array with a depth of 1 μm. Additionally, there are still Bi_2_Se_3_ films on top of the Si NWs. The optical properties of Si NWs and Bi_2_Se_3_/Si NWs were also measured, as shown in Figure 4e,f. From Figure 4e, it can be observed that the reflectance of Si NWs is ~10%, far lower than that of ordinary Si wafers [49]. This verifies that the Si NWs has a good antireflection function in the wavelength range of 350 nm to 1000 nm. However, it increased to be >10% after the deposition of the Bi_2_Se_3_ film; at wavelength >1000 nm, the reflectance of Si NWs suddenly increased to ~60% due to its non-absorption, which is similar with that of the Si wafer [50]. The reflectance of the Bi_2_Se_3_ film was about 10~20% in the measurement rang. Figure 4f shows that Si NWs can absorb light at a wavelength <1000 nm, but Bi_2_Se_3_/Si NWs can absorb light up to 2500 nm due to the direct bandgap of 0.3 eV (with an absorption edge of 4.1 μm).

Following the covering of the Bi_2_Se_3_ film on Si NWs, we can observe that the reflectance increased because the deposited Bi_2_Se_3_ entered the interspace between the Si NWs, which prevented part of the light from entering the Si NWs, and thus reduced the antireflection function. In the wavelength >1100 nm, the reflectance decreased considerably from 60% for the Si NWs to 10% for the Bi_2_Se_3_/Si NWs, due to the much smaller bandgap (0.3 eV) of Bi_2_Se_3_ than that (1.1 eV) of silicon. Figure 4f shows that, when the wavelength is shorter than 1000 nm, both Si NWs and Bi_2_Se_3_/Si NWs have a transmittance of 0 since both of them have a bandgap smaller than 1.1 eV (corresponding to an absorption edge of 1100 nm), and thus the incident light is completely absorbed. At the wavelength >1100 nm, the transmittance in the Si NWs is ~30%, but in the Bi_2_Se_3_/Si NWs sample it is only less than 10%. The reason is that the Bi_2_Se_3_ has a bandgap of 0.3 eV and so it can absorb the light with a wavelength shorter than 4.1 μm in theory.

### 3.4. Performance of the n-Bi_2_Se_3_/p-Si NWs Photodetector

The n-Bi_2_Se_3_/p-Si NWs sample was covered by an Ag interdigital electrode at the front side and by the fully covered Ag film at its back side, as shown in Figure 5a. Then, a simple photodetector was completed. Figure 5 shows the performance of a typical device, and the whole fabrication process is schematically shown in Figure 5a. Figure 5b shows the responsivity of the n-Bi_2_Se_3_/p-Si NWs photodetector at the wavelength range of 390 nm to 1700 nm. The highest responsivity is 84.3 mA/W at the initial wavelength of 390 nm. Then, it drops linearly to 2.68 mA/W at 528 nm. Following this, it rises to 14.17 mA/W at 960 nm and drops again to 1.15 mA/W at 1200 nm. Subsequently, it increases again to 7 mA/W at 1700 nm. Figure 5c shows the detectivity of the n-Bi_2_Se_3_/p-Si NWs photodetector at the wavelength range of 390 nm to 1700 nm. The largest detectivity is 1.06 × 10^10^ Jones at the initial wavelength of 390 nm. Then, it drops linearly to 3.35 × 10^8^ Jones at 528 nm. Subsequently, it rises to 1.79 × 10^9^ Jones at 960 nm and drops again to 1.44 × 10^8^ Jones at 1200 nm. Then, it increases again to 8.28 × 10^8^ Jones at 1700 nm. In order to explain the complex change of responsivity, we drew the absorption coefficient and absorption depth of Bi_2_Se_3_ and Si in this wavelength range [24,25], as shown in Appendix A. In the wavelength range of 400 to 1700 nm, the absorption coefficient of Bi_2_Se_3_ was between 10^6^ to 10^5^ cm^−1^, while that of Si rapidly dropped from 10^4^ at ~400 nm to 10^−8^ at ~1500 nm. Additionally, in view of the cross-section morphology shown in Figure 4c,d, we considered that the change in responsivity can be explained as follows: (1) at ~400 nm, the light is mainly absorbed by the surface Bi_2_Se_3_ compact film. The photo-generated electrons can be transported efficiently to the Ag front electrode through the surface Bi_2_Se_3_ compact film, while the photo-generated holes can be transported efficiently to the Ag back electrode through Si NWs and the Si substrate, as shown by the energy-band diagrams shown in Appendix A. Therefore, the responsivity is high. (2) From ~400 nm to ~528 nm, the reflectivity of the Bi_2_Se_3_/Si NWs sample increased, as shown in Figure 4e; the surface Bi_2_Se_3_ compact film cannot absorb the incident light enough and the leaky light is absorbed by the loose Bi_2_Se_3_ film inside the Si NWs interspace. In this situation, the transportation of the photo-generated carriers to the Ag electrode is limited due to the loose contact between the Bi_2_Se_3_ nanosheets and between them with Si NWs. (3) From ~528 nm to ~1200 nm, the leaky part of the incident light begins to be absorbed by the Si substrate. The photo-generated electrons diffuse through Si NWs and drift through Bi_2_Se_3_ to reach the Ag front electrode, while the photo-generated holes are efficiently transported to the Ag back electrode. The responsivity reaches the local largest value of 14.17 mA/W. (4) From ~1200 nm to 1700 nm, the light can only be absorbed by the Bi_2_Se_3_ film and the responsivity increases again. Therefore, we believe that part of the absorption by Si NWs does not contribute to the light current. This can be easily understood because there are many broken Si NWs (see Figure 4d) that do not make contact with the above Bi_2_Se_3_ film, and so the produced carrier cannot be transported to the front electrode.

Figure 5d shows the transient photocurrent measured on the n-Bi_2_Se_3_/p-Si NWs photodetector under the illumination of a 980 nm laser with a power of 0.6 W. The forward voltage is 1 V. Its rise time is 3 ms and its fall time is 1 ms. In view of the long diffusion length of the carrier, this response is very rapid, which can mainly be attributed to the function of bulk heterostructure. Figure 5e shows its photovoltaic behavior under the illumination of a 980 nm laser (0.6 W). The power density of the laser was about 92.3 W/cm^2^. As shown in Figure 5e, V_oc_ is 0.034 V, J_sc_ is 4.19 mA/cm^2^ and FF is 23% yielding a power conversion efficiency (PCE) of 3.59 × 10^−5^%. These results imply that our n-Bi_2_Se_3_/p-Si photodetector can work in a self-powered mode with a zero external bias voltage, although its PCE is still very small at present. 

We summarize the performances of Bi_2_Se_3_/Si photodetectors previously reported by others and ourselves in Appendix A. For the type of Bi_2_Se_3_/p-Si photodetector, our detector has the longest detection wavelength and the fastest rise and fall times than those reported by and M. Li et al. [19]. We considered that the longer detection wavelength can be attributed to the strong IR absorption (~75%, see Figure 4e,f) for our sample that came from the closely assembled Bi_2_Se_3_ film, taking advantage of the vacuum evaporation method. The rapid rise and fall times can be ascribed to the bulk heterojunction that can facilitate the dissociating of photo-generated carriers and their transportation to the metal electrode. In comparison to the performance of the Bi_2_Se_3_/n-Si photodetector, our photodetector had a longer detection wavelength than that reported by C. Liu et al. [18] and H. Zhang et al. [16], but was shorter than that reported by X. Hong et al. [17]. However, the responsivity of our photodetector was 10^5^ times better than theirs at the wavelength of ~1.5 µm, which was due to our Bi_2_Se_3_ film that was thicker than theirs. It is interesting that our Bi_2_Se_3_/p-Si photodetector had a very high responsivity at 390 nm, which has not been observed previously by others. We considered that one high-energy ultraviolet (UV) photon may pump out more than one exciton inside the Bi_2_Se_3_, which indicated that Bi_2_Se_3_ could be an excellent material for a UV photodetector. For the typical 2D materials, MoS_2_ [51] and reduced graphene oxide (RGO) [52], they were also deposited onto Si or Si NWs, respectively, to fabricate the photodetectors. The detection wavelength of the former was only up to 1050 nm, much shorter than ours. Although the RGO/Si NWs photodetector can respond up to 10.6 μm, it cannot work in a self-powered mode. In general, the performance of our primitive n-Bi_2_Se_3_/p-Si NWs photodetector is comparative to others at present. Many studies on the morphology of Bi_2_Se_3_ films and photodetector structure are ongoing to further improve their performance.

## 4. Conclusions

We obtained high-quality topological insulator Bi_2_Se_3_ thin film by the vacuum thermal evaporation method, and analyzed their phase structure, chemical composition and surface morphology, and also obtained the best growth conditions after optimizing the temperatures of Bi and Se source cells, as well as the growth temperature. Then, PPMS was used to analyze the electrical properties of the Bi_2_Se_3_ thin film. We found that the Bi_2_Se_3_ film had a metallic state, WAL and LMR characteristics, typical for the TI material. However, these topological properties were measured at low temperatures (<10 K) and cannot be directly related to the performance of the detector measured at room temperature. Then, Si NWs was prepared by the MACE method and its interspace and height were tuned until its morphology was most suitable for the n-Bi_2_Se_3_/p-Si NWs photodetector. It was covered by a thick Bi_2_Se_3_ film deposited by the vacuum thermal evaporation method to form the bulk heterojunction photodetectors for the first time. Finally, we obtained a self-powered n-Bi_2_Se_3_/p-Si NWs photodetector with the best performance in our case: its detection wavelength ranged from 390 nm to 1700 nm, its highest responsivity was 84.3 mA/W at 390 nm and its response time reached 3 ms/1 ms, which can be attributed to the bulk heterostructure. Research efforts are underway to modify the Bi_2_Se_3_ filling inside Si NWs and its crystal shape as well as device structure to improve the photodetector’s performance. Since the topological properties of the Bi_2_Se_3_ film were only observed at low temperatures, some excellent properties of the n-Bi_2_Se_3_/p-Si NWs photodetector obtained at room temperature should only be related to the common semiconductor properties of the Bi_2_Se_3_ film. In order to verify the effect of topological properties on the photodetector’s performance, topological and non-topological Bi_2_Se_3_ films should be deposited onto Si NWs to form a photodetector and their performance should be measure at low temperatures (<10 K).

## Figures and Tables

**Figure 1 nanomaterials-12-01824-f001:**
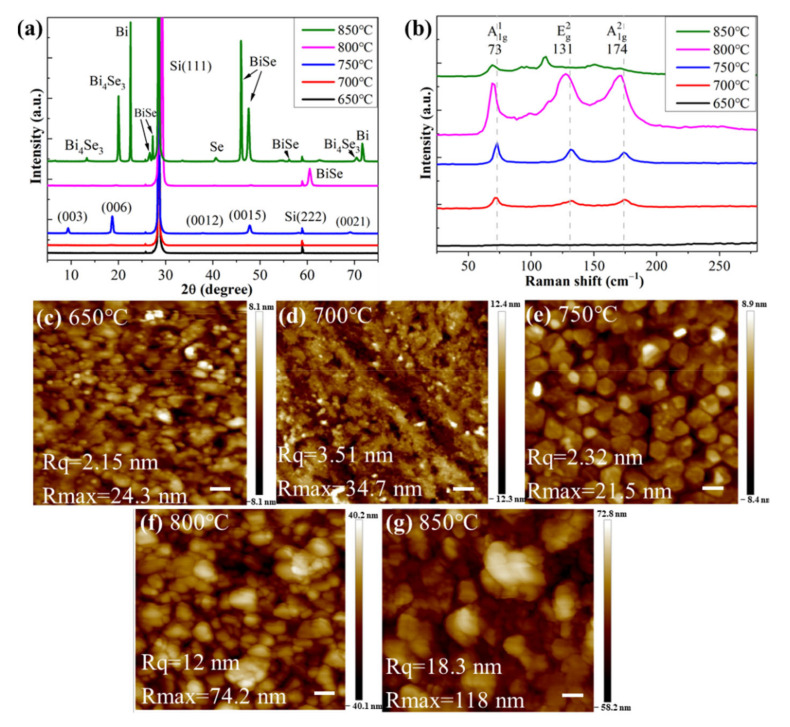
Characterization of Bi_2_Se_3_ films grown at different Bi cell temperatures (650~850 °C) and at the same growth temperature of 320 °C and Se source cell temperature of 209 °C: (**a**) XRD; (**b**) Raman; (**c**–**g**) AFM of sample surface. Scale bars in (**c**–**g**) are all 100 nm.

**Figure 2 nanomaterials-12-01824-f002:**
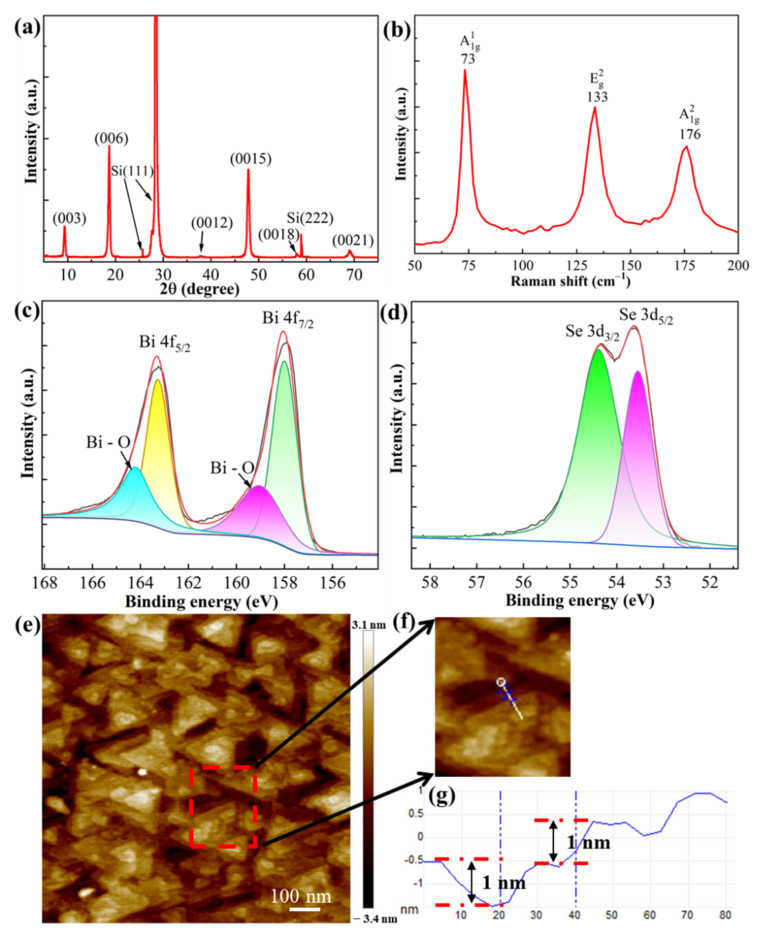
Bi_2_Se_3_ film grown at the optimal condition: Bi cell temperature of 750 °C, Se cell temperature of 204 °C and growth temperature of 320 °C. (**a**) XRD; (**b**) Raman; (**c**,**d**) XPS of Bi 4f and Se 3d with the fitting curves; (**e**) AFM of Bi_2_Se_3_ film surface and (**f**) local enlargement; (**g**) atomic step height.

**Figure 3 nanomaterials-12-01824-f003:**
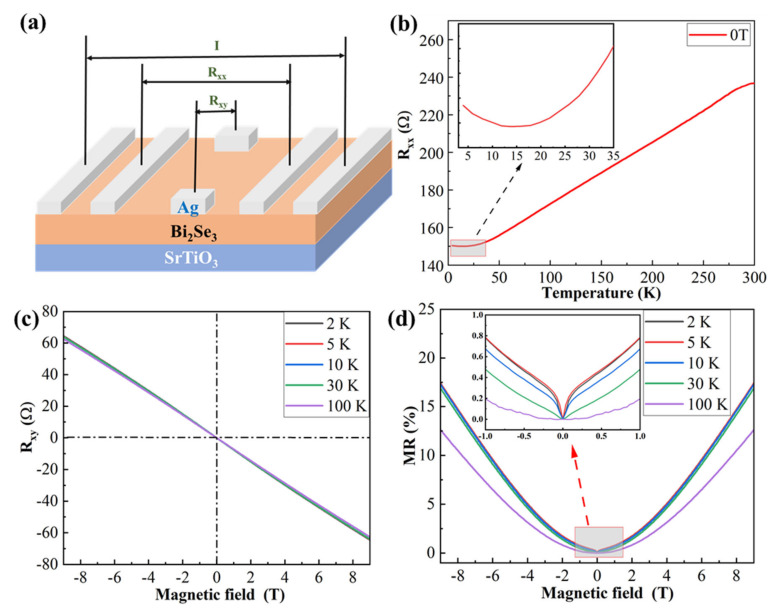
Electrical transport measurement of Bi_2_Se_3_ thin film. (**a**) Schematic diagram of sample and electrodes. (**b**) R_xx_ curves as a function of temperature under magnetic fields of 0 T and 9 T (the inset is an enlarged view at <35 K); (**c**) Hall resistance curves as a function of magnetic field at different temperatures; (**d**) magnetoresistance curve of the sample as a function of magnetic field (the inset is an enlarged view of the low magnetic field −1~1 T).

**Figure 4 nanomaterials-12-01824-f004:**
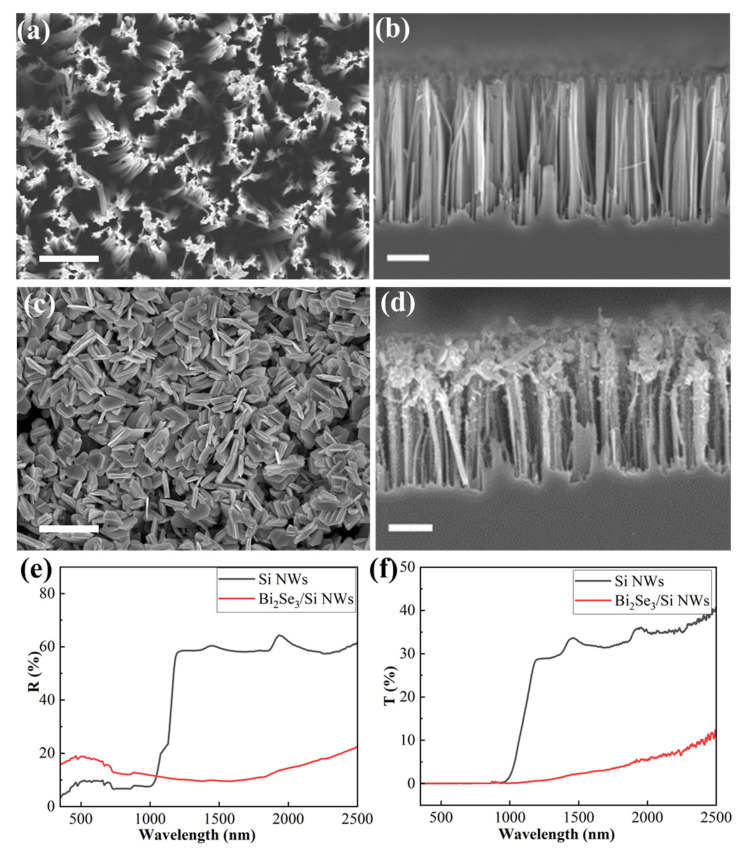
(**a**) Surface and (**b**) cross-section SEM images of Si NWs with the deposition time of 60 s for Ag NPs and the etching time of 10 min; (**c**) surface and (**d**) cross-section SEM images of Bi_2_Se_3_/Si NWs heterojunction with the deposition time of 60 s for Ag NPs and the etching time of 10 min; (**e**) reflectance and (**f**) transmittance of Si NWs and Bi_2_Se_3_/Si NWs samples. Scale bars in (**a**–**d**) are all 1 μm.

**Figure 5 nanomaterials-12-01824-f005:**
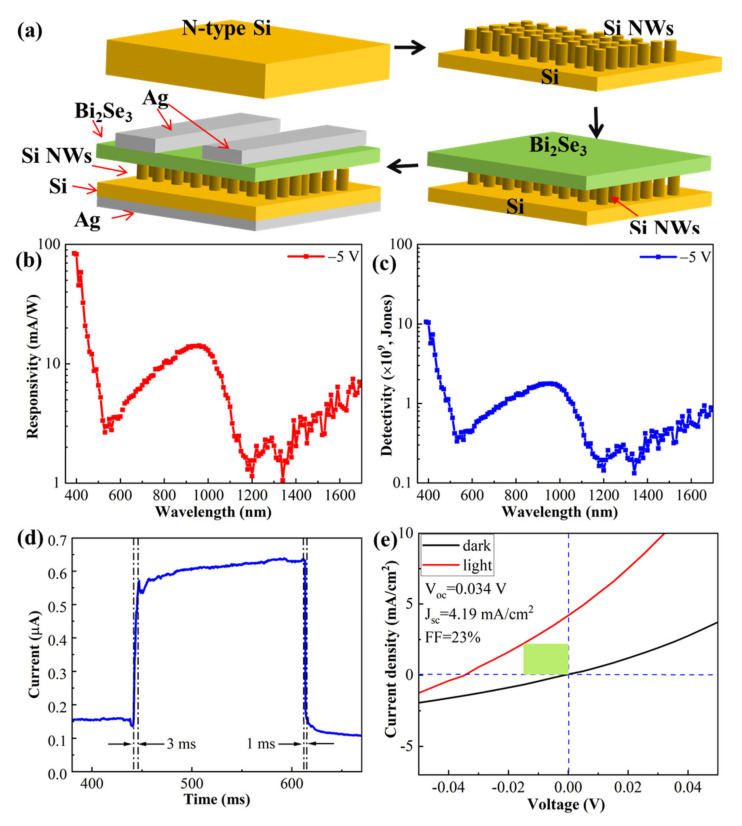
Characterization of n-Bi_2_Se_3_/p-Si photodetector: (**a**) its fabrication process; (**b**) its responsivity and (**c**) its detectivity at the wavelength range from 390 nm to 1700 nm; (**d**) its response times and (**e**) its photovoltaic behavior measured under a 980 nm laser illumination with a laser power of 0.6 W.

## Data Availability

Data are contained within the article.

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
