# Peer review of "Fabrication and Characterization of a Self-Powered n-Bi2Se3/p-Si Nanowire Bulk Heterojunction Broadband Photodetector"

_nanomaterials, 2022, doi:10.3390/nano12111824_

Round 1

Reviewer 1 Report

Summary

 The authors report fabrication and characterization of a BiS3 / Si nanowire heterojunction photodetector with broadband response from 390nm to 1700 nm. They report studies optimizing the growth of the BiSe films and performance electrical characterization to illustrate the topological insulator properties of these materials at low-temperatures. Their main findings include the growth process for these materials by vacuum evaporation and the formation of the bulk heterojunction by the etching of Si NW features. They claim that the devices have an impressive responsivity and fast response speed for NIR and mid IR photodetection.

General Comments:

  • The full manuscript never pasted in the abstract into the template. The abstract is shown in the review system, but not in the manuscript. The abstract should also so as to introduce themes of the manuscript but not list all of the numerical details, for example, of the precursor temperatures during the growth process. These details are not appropriate for an abstract that would encourage a broad readership.
  • The ‘WAL’ effect is not spelled out in the introduction – it should be clear what this stands for before the acronym is used
  • It would be a good idea to briefly compare the performance of BiSe photodetectors with the rest of the field of photodetectors in the introduction. It is interesting that it can also be a topological insulator, but does this enhance performance beyond the state of the art?
  • It would also be helpful to clarify whether the processes for growth of BiSe can be suitable for on-chip integration and heterogeneous integration with silicon integrated circuits? Is there any potential for low temperature growth to allow this integration?
  • Is the vacuum evaporation method for BiSe deposition part of the novelty or has this process been done before? It should be clear from the abstract and introduction which parts of this work are unique and innovative.
  • Figure 1 and Figure 2 should show scale bars on the AFM images. Also, the text in the labels on these figures is too small to read
  • Figures 1-3 feel a bit repetitive. The reader may at first glance think that they are replicates. These figures should be condensed or some of the data should be moved to the supporting information. There are too many similar images and plots here to digest.
  • It would be helpful to connect the properties such as the LMR observed in Figure 5 to the photodetector performance. It should be made clear to the reader that these properties are important and relevant to the device application being developed
  • The authors can improve this work by making more rigorous, quantitative comparisons to the scientific literature and the state of the art. They claim the high performance of their new device architecture, but it is not necessarily clear what the comparison point is for this claim? There should be some greater justification for this material and device architecture beyond a comparison simply against other BiS3 photodetectors. If this is the only claim, then the work may appear more incremental. The nanowire heterojuction structure is more complex. Does its performance warrant the additional fabrication steps and cost?

Figures

  • Several figures have the labels ‘(a)’, etc. located too close to the figures’ axes, in several cases, overlapping.

Recommendation: This manuscript requires significant revisions to become publishable. I recommend revision and resubmission. With further clarification of the unique and innovative aspects of this work, condensing of Figures 1-3, and improvements to the writing, it could become an interesting paper for a broader audience. The authors can improve this work by making more rigorous comparisons to the scientific literature and directly comparing device metrics with the state of the art. Another feature which can strengthen this work is if the authors can give more interesting discussion that directly relates to the topological insulator properties with the device performance. It should be clear that the properties are, in fact, beneficial for enhancing the photodetection.

Reviewer 2 Report

The authors studied well about a photodetector with a nanowire structure. However, there is a missing essential part, such as the abstract, and English should be improved. Professional proofreading is required. 

1. In Figure 7, It is more helpful to show more broad J-V characteristics, over 1V, since photodetector typically works under negative bias. 

2. How about the surface roughness of Bi2Se3 on Si NWs? Are there any short issues after Ag deposition due to the high surface roughness of Si NW? 

3. How about specific detectivity of the Bi2Se3/Si photodetector? 

Reviewer 3 Report

In this article the authors synthesized high quality topological insulator Bi2Se3 thin film by vacuum thermal evaporation and analyzed their various properties to optimize best growth conditions. After optimizing the best growth conditions, they studied the performance of that photodetector. Although the manuscript is well written there is still so many short comings which should be addressed before publication

  1. The abstract is not written related to the manuscript. Authors should add clear and precise abstract in the abstract section.

  1. The title of the manuscript is related to photodetector but in the main manuscript mostly focus is about the optimization process of the Bi2Se3/Si NWs. In the manuscript, it seems that mainly the work is related to the optimization for the best growth parameters and the at the end they have just tested it for the photodetector application.

  1. In the Introduction section the authors should focus on the motivation for this study. Why is this study required?

  1. The materials and method section can be improved by adding more details so other researchers can reproduce the results. Like what thickness of Ag films is being used and what are the process conditions for magnetron sputtering?

  1. In section 3. Results and discussion line 128 and 129 the author writes that “The XRD, Raman, and AFM … are shown in figure S1. These all details are not in Figure S1 instead they are in Figure 1. Please check and correct it accordingly. Similarly, while explaining the XRD results the authors used the term S1(a) which is not correct.

  1. In section 3. Results and discussion line 142 the vibrational mode A1g is written 1g should be in subscript.

  1. In section 3. Results and discussion line 149 to 151 the author states “For Bi cell temperature at 700 … and by this they conclude that the deposited films are thin, and the deposition rate is lower. My question is can you please comment that what is the effect of temperature on rate of deposition and why?

  1. Why you choose these specific temperature ranges for Bi and Se cells? Is there any other literature available from which you get the idea for these specific temperature ranges?

  1. In section 3. Results and discussion line 189 the authors claim that the growth temperature was changed from 250 to 450 with interval of 50. But in the results presented in figure 3 and its explanation shows that the growth temperature change is from 220 to 420. Please check.

  1. In the caption of figure 4. The caption for figure 4 (e) is missing.

  1. In section 3. Results and discussion line 210. The authors relate the peaks present at binding energy with figure 1f. My question is how they are related please explain.

  1. In section 3. Results and discussion line 278-283. By doing the things mentioned in those lines what results author gets and what they concluded from those results. Please explain in detail.

  1. In section 3. Results and discussion line 284 the authors claim that Ag films electrodes are shown in figure 8(a). But in the manuscript, there are only 7 figures. There is no figure 8. Please check this.

  1. The explanation of optical properties is so confusing and is bizarre. The authors only mention the results which can be seen in the figure as well. Please explain the reasons clearly and try to link them with some literature.

  1. In Figure 7 the figure labelled as (e) is not in the explanation that should be labelled as (d).

  1. The authors should clearly state the novelty of the work in introduction and conclusion section.

  1. The authors tested and presented results of photodetector for only one device. The conclusions are too far-fetched considering only one device is tested. What about the reproducibility of the device?

  1. Figure 7 (c) y-axis unit should be revised.
  2. In Figure 7b, at what bias photoresponsivity is measured?
  3. In introduction, authors should discuss the typically TMDCs based photodetectors to compare with Bi2Se3. So, it better to discuss more articles in this section like: (a) https://pubs.rsc.org/en/content/articlelanding/2020/nr/d0nr05737a. (b) https://www.sciencedirect.com/science/article/abs/pii/S0169433221001501
  4. There several typos and English must be improved.

Round 2

Reviewer 1 Report

The authors have fixed most of the issues with this manuscript that were pointed out in the initial review. There are only a few outstanding issues to correct. We also encourage the authors to perform detailed editting for small English errors (omission of definite articles, etc.). These edits will improve the readability of their work and lead to more citations. We also suggest a few minor changes to improve the clarity of the figures.

Top of Form

1) in response to Reviewer 1, comment #6, the authors have added scale bars to Figure 1. The first scale bar has a length labeled but the others are not labeled. Best practice is either to label all scale bars in the figure itself, or to remove the labels and specify them in the figure caption if they are all the same. Otherwise, it could be ambiguous for the reader. Same goes for the new supporting information figures.

2) Along the same lines, I suggest the authors either resize or reposition the scale bar for Figure 4a, since it is a bit hard to see because of the poor contrast with the image behind it. Figure 4c could also be repositioned. Putting the scalebar text into the caption fixes part of this issue.

3) For figure 5d, please be sure that all of the specifications for the rise / fall time experiment are specified, for example, the wavelength and power used for illumination with the pulse

4) The discussion of the self-powered property on pg. 18 should mention what the power is per area at a nominal incident intensity (make sure to list the assumption). This is important for noting whether the self powering function will be very impactful. This is also important because although self-powered is part of the title of this work, it does not receive much attention in the text or the abstract / conclusion. I suggest another mention of this aspect or the reader may feel that it was never discussed.

Reviewer 2 Report

The authors studied about the photodetector based on Bi2Se3/p-Si nanowires. The article is well organized and acceptable for publication.

Author Response

Thank you very much for your great efforts!

Reviewer 3 Report

The authors revised the manuscript adequately. Now I feel it should be published in this journal.

Author Response

(The authors gave the same response as above.)
